# Effect of Cavitation Erosion Wear, Vibration Tumbling, and Heat Treatment on Additively Manufactured Surface Quality and Properties

**Sergey N. Grigoriev** [1], **Alexander S. Metel** [1], **Tatiana V. Tarasova** [1],
**Anastasia A. Filatova** [1,*], **Sergey K. Sundukov** [2], **Marina A. Volosova** [1],
**Anna A. Okunkova** [1,*], **Yury A. Melnik** [1] and **Pavel A. Podrabinnik** [1]

[1] Department of High-Efficiency Processing Technologies, Moscow State University of Technology «STANKIN», Vadkovsky per. 1, 127055 Moscow, Russia; s.grigoriev@stankin.ru (S.N.G.); a.metel@stankin.ru (A.S.M.); t.tarasova@stankin.ru (T.V.T.); m.volosova@stankin.ru (M.A.V.); yu.melnik@stankin.ru (Y.A.M.); p.podrabinnik@stankin.ru (P.A.P.)

[2] Department "Technology of Construction Materials", Moscow Automobile and Road Construction State Technical University (MADI), Leningradsky Prospect 64, 125319 Moscow, Russia; s.sundukov@madi.ru

[*] Correspondence: a.filatova@stankin.ru (A.A.F.); a.okunkova@stankin.ru (A.A.O.); Tel.: +7-909-913-1207 (A.A.O.)

**Abstract:** The paper is devoted to researching various post-processing methods that affect surface quality, physical properties, and mechanical properties of laser additively manufactured steel parts. The samples made of two types of anticorrosion steels—20kH13 (DIN 1.4021, X20Cr13, AISI 420) and 12kH18N9T (DIN 1.4541, X10CrNiTi18-10, AISI 321) steels—of martensitic and austenitic class were subjected to cavitation abrasive finishing and vibration tumbling. The roughness parameter $R_a$ was reduced by 4.2 times for the 20kH13 (X20Cr13) sample by cavitation-abrasive finishing when the roughness parameter $R_a$ for 12kH18N9T (X10CrNiTi18-10) sample was reduced by 2.8 times by vibratory tumbling. The factors of cavitation-abrasive finishing were quantitatively evaluated and mathematically supported. The samples after low tempering at 240 °C in air, at 680 °C in oil, and annealing at 760 °C in air were compared with cast samples after quenching at 1030 °C and tempering at 240 °C in air, 680 °C in oil. It was shown that the strength characteristics increased by ~15% for 20kH13 (X20Cr13) steel and ~20% for 12kH18N9T (X10CrNiTi18-10) steel than for traditionally heat-treated cast samples. The wear resistance of 20kH13 (X20Cr13) steel during abrasive wear correlated with measured hardness and decreased with an increase in tempering temperatures.

**Keywords:** anticorrosion steel; hardness; laser powder bed fusion; microroughness; tensile test; wear resistance

## 1. Introduction

Analysis of modern aircraft designs shows that about 50%–60% of the parts that form the outer contour of the product and parts of the inner set can be manufactured in monolithic structures by various methods of additive manufacturing [1–5]. The labor intensity of manufacturing parts from metal powders under conditions of mass production will not exceed 35%–60% of the total labor intensity of their production from deformable materials in the form of prefabricated structures, which creates a significant economic effect [6].

Such laser powder bed fusion factors like laser power, beam spot size, laser beam profile, scanning speed, strategy and hatch distance, powder particle shape, and morphology characteristics (bulk density of the powder in a layer that also depends on the way of leveling), in combination, affect not only

surface quality, but also the main physical, mechanical, and exploitation properties [7–11]. Regardless of all the research attempts optimizing the factors, the surface quality of the produced parts and physical, mechanical, and exploitation properties stayed under the required level for the real industrial applications and were reported multiple times [12–16].

The main problem of surface quality is the natural waviness of the produced surface and unmelted granules trapped in the molten pool [17–21] that are especially important for inner cavities of complex geometrical products since it can reduce the functionality of the surfaces—their wear resistance in friction pairs that strongly depend on submicron roughness [22–25]. The impossibility of polishing inner and complex-profile areas by most known and widespread post-processing methods hampers additive manufacturing's widespread and consequent transfer to the sixth technological paradigm [26–29] that determines the relevance of developing scientific and technological principles of finishing of the parts obtained with the laser powder bed fusion method.

One of the most popular is mechanical polishing, which strongly depends on the size of the used abrasive, retains quite typical traces of abrasive wear, and provides one of the best polishing effects, but makes it impossible for the application to the inner cavities of the complex shaped parts of real production [30–32]. Creation of the specified polishing tools to achieve inner cavities handsomely hampered by the typical irregular topology of obtained surfaces, has difficulties in control of even geometry, and remains a labor-intensive task [14–16,33]. The method of laser-plasma polishing occurs in a metal vapor that prevents oxidation, has the local impact of the laser beam of a relatively small laser spot [34–36], has a severe problem similar to the mechanical polishing methods related to the linear nature of coherent light propagation that is blocked for inner cavities by the geometry of the complex part. The same problem can be detected during high-current pulsed electron beams polishing, allowing almost the eliminating of porosity and reducing the roughness parameter $R_a$ from tens to several micrometers [37–39]. Besides, it does not reduce the roughness parameter $R_a$ of less than 1 μm when actual mechanical polishing allows the roughness parameter $R_a$ reduction up to approximately 0.04 μm that corresponds to the highest class of surface cleanliness [40,41].

Electrochemical etching is of the disadvantages of beam methods, allowing the finishing of complex parts up to the roughness parameter $R_a$ of ~0.04 μm. However, using a specified electrolyte for each material has a potential threat to the environment and hampers its widespread use for additive manufactured parts, which remains one of the most meaningful methods for a large field of applications [42,43].

One of the most promising post-processing methods for the surface treatment of complex-shaped parts remains underestimated—processing in a gas discharge plasma [44–46]. It is free of the inability to process inner cavities and channels of the part with the most sophisticated geometry. Explosive ablation of surface protrusions, polishing with a concentrated beam of fast neutral argon atoms at a large angle of incidence, surface coating deposition upon sputtering with argon ions of solid magnetron targets, and/or evaporation of a liquid metal magnetron target heated by ions allows reduction of the roughness parameter $R_a$ to 0.1–0.2 μm with a decrease in pulse width to 1.5 ns. It cannot be considered an example of protrusion removal on the part of the surface immersed in a plasma due to explosive ablation when high-voltage pulses are applied. However, there are still a few works searching for the method's full potential for additive manufacturing.

Ultrasonic and vibratory finishing is known as mechanical surface treatment methods based on the complex mechanical nature of the action and is considered as a traditional alternative for post-processing methods requiring the setup of a sophisticated unit [47–52], that can be a strong preference in the conditions of aircraft part production. Another advantage is processing more large-scale functional parts with an overall size of more than 100 mm. An important feature of ultrasonic liquid finishing is that the working bodies are cavitation cavities that arise in a liquid under the action of ultrasonic vibrations, which makes it possible to process surfaces of any complexity. At present, there are no actual results of the experiments on the successful use of ultrasonic treatment to reduce the roughness of laser additively manufactured parts, developed recommendations, and strong

mathematical support, especially for the parts made of structural anti-corrosion steels of austenitic and martensitic class that stay traditionally under the demand of the real production.

The same problem is related to the research of heat treatment effect on the physical and mechanical properties of the laser additively manufactured parts produced from anti-corrosion steel and their effect on the samples' wear resistance since most of the work is devoted to the quite well-known cast parts [53,54]. However, laser additively manufactured steel parts have other problems. Since they were already heat-treated and remelted multiple times with a laser beam, they should not be additionally quenched to improve their hardness, but they remain with the strong anisotropy of the properties that can be reduced with a developed complex of the post-treatment based on traditional approaches (low tempering at 240 °C in air, at 680 °C in oil, annealing at 760 °C in air) that needs an experimental approval.

In this regard, the work investigated the prospects for using ultrasound post-processing methods to improve the surface quality parameters, topology and the effect of various heat-treatment methods on the physical and mechanical properties of produced samples compared to the cast parts' wear resistance.

The scientific novelty of the work is in researching post-processing methods, including heat treatment (tempering and annealing) and polishing methods based on mechanical nature (ultrasonic cavitation finishing and vibratory tumbling) and their modes, including mathematical support, for additively manufactured parts from corrosion-resistance steels for the aircraft industry and their influence on mechanical properties and surface roughness of the complex-shaped parts.

The purpose of the work is to determine the effect of post-processing modes on the hardness, resistance to abrasive wear, surface roughness parameters (arithmetic mean deviation ($R_a$), ten-point height ($R_z$), and maximum peak-to-valley height ($R_{tm}$)) of additively manufactured parts produced by the laser powder bed fusion method to ensure the required properties of aircraft parts made of corrosion-resistant steels of austenitic and martensitic classes.

The results obtained for 20kH13 (DIN 1.4021, X20Cr13, AISI 420) steel are required for a quarter-turn lock mechanism of the aircraft that includes a pin, washer, and sleeve. Since the parts of the lock mechanism are with a diameter of 11 mm, a height of 7 mm, and are complex shaped, the traditional production route is rather laborious and complicated when the application of the laser powder bed fusion method for its production simplifies the operational way without loss of the part exploitation properties. The material of the washer should be wear-resistant since there are friction surfaces between the two parts. The material should differ in strength from the pin material by 20%, which should provide no sticking effect between the parts; the strength of the lock pin is not less than 1300 MPa. The required hardness is not less than 42 HRC, the density is not less than 7.7 g·cm$^{-3}$, and roughness parameter $R_a$ is less than 3.2 μm.

Another airplane part made of 12kH18N9T (DIN 1.4541, X10CrNiTi18-10, AISI 321) that requires experimental data is an air intake grille module steel, which is an responsible element for protecting the air intake duct from the objects entering it and is an obstacle to the air intake to the engine with overall dimensions of 180 mm × 100 mm × 30 mm with the minimal thickness of the inclined wall of 0.3 mm. The part should be produced following quality requirements: tensile strength is not less than for a standard semi-finished product with a density of less than 7.9 g·cm$^{-3}$, roughness parameter $R_a$ of less than 6.3 μm. The traditionally produced grille modules are characterized by significant labor intensity. Their manufacturing path includes many operational steps—cutting, bending, manual assembly of almost seventy parts, welding, soldering, etc. Its direct laser manufacturing from the powder takes the production to a new level [55–58].

## 2. Materials and Methods

### 2.1. Materials

A wide range of aircraft parts are manufactured from corrosion-resistant steels of the martensitic and austenitic classes that provide the required exploitation properties. The researching of post-processing

methods and modes were conducted with samples made of corrosion-resistant steel of the martensitic class, grade 20kH13 (analog DIN 1.4021, X20Cr13 according to EN 10088-4), and corrosion-resistant chromium-nickel steel of the austenitic class, grade 12kH18N9T (analog DIN 1.4541, X10CrNiTi18-10 according to EN 10088-1) under the request of the aviation industry enterprise. The chemical composition is presented in Table 1. PR 20kH13 powder with a fraction of 40 μm and PR 12kH18N9T powder with 20 to 63 μm (JSC POLEMA, Tula, Russia) were manufactured by dispersing molten metal with a jet of compressed gas [59,60].

**Table 1.** Chemical composition of the steel powder grades PR 20kH13 and PR 12kH18N9T [%].

| Material | C | S | P | Mn | Cr | W | V | Si | Ni | Mo | Cu |
|---|---|---|---|---|---|---|---|---|---|---|---|
| PR 20kH13 | 0.16–0.25 | ≤0.025 | ≤0.03 | ≤0.6 | 12–14 | - | - | ≤0.6 | ≤0.6 | - | - |
| PR 12kH18N9T | ≤0.12 | ≤0.02 | ≤0.035 | ≤2.0 | 17–19 | ≤0.2 | ≤0.2 | ≤0.8 | 8–9.5 | ≤0.5 | ≤0.4 |

## 2.2. Equipment

Laser powder bed fusion was carried out on an EOS M280 industrial unit (EOS GmbH, Krailling, Germany) and an ALAM experimental laser powder bed fusion setup (MSTU Stankin, Moscow, Russia) [61,62] equipped with the laser source of continuous radiation LK-200 (IPG LASER GMBH, Fryazino, Russia), equipped with two attenuators, B-Cube and C-Varm (Coherent, Santa Clara, CA, USA), and a Focal-piShaper (πShaper, Berlin, Germany). The experimental unit had the following technical parameters: a wavelength of 1070 nm, a beam divergence of 0.2°, maximum power $P_{l.max}$ of 200 W (Figure 1).

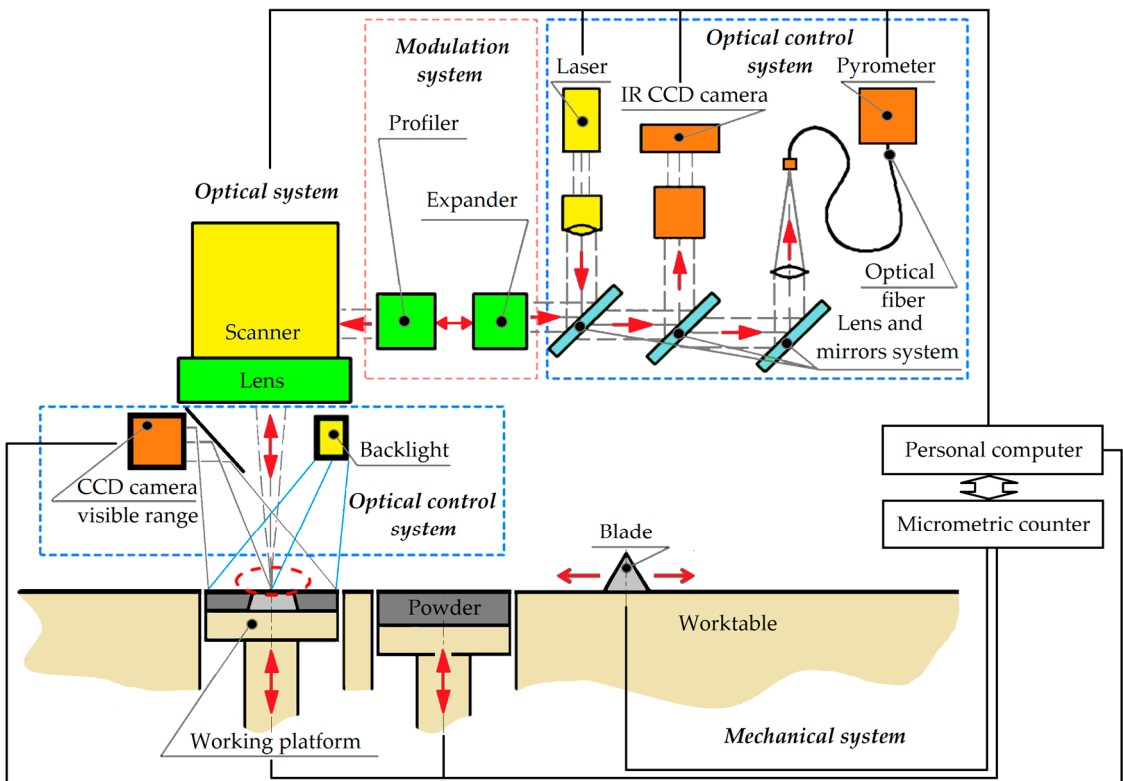

**Figure 1.** Experimental setup, where *CCD* is a charge-coupled device, *IR* is infrared.

The powders were sifted using an analytical sieving machine AS200 basic (Retsch, Dusseldorf, Germany) with test sieves of 40 μm by ISO 3310-1 and dried by a vacuum oven VO400 (Memmert GmbH + Co. KG, Schwabach, Germany) before processing. The powder drying removed the excess air and contributed to a better powder density in the layers.

A parametric analysis of experimental work was carried out on the ALAM and EOS M 280 setup to select the optimal modes' manufacturing samples.

The chosen laser powder bed fusion factors at the EOS M280 and ALAM setups for growing solids of 20 mm × 20 mm × 20 mm samples for five pieces for each of experimental set are presented in Table 2. For tensile tests, 15 flat specimens were used in accordance with GOST 11,701 (Figure 2). The specimens were grown in the direction of the Z axis, the angle of inclination of the longitudinal axis of the specimen to the printing plane was 0°.

**Table 2.** Laser powder bed fusion factors for growing solids. 10 ± 0.1.

| Material | Layer Thickness (µm) | Laser Radiation Power $P_l$ (W) | Scanning Speed vs. (mm·s$^{-1}$) |
|---|---|---|---|
| PR 20kH13 (AISI 420) | 20 | 80 | 390 |
| PR 12kH18N9T (AISI 321) | 20 | 100 | 100 |

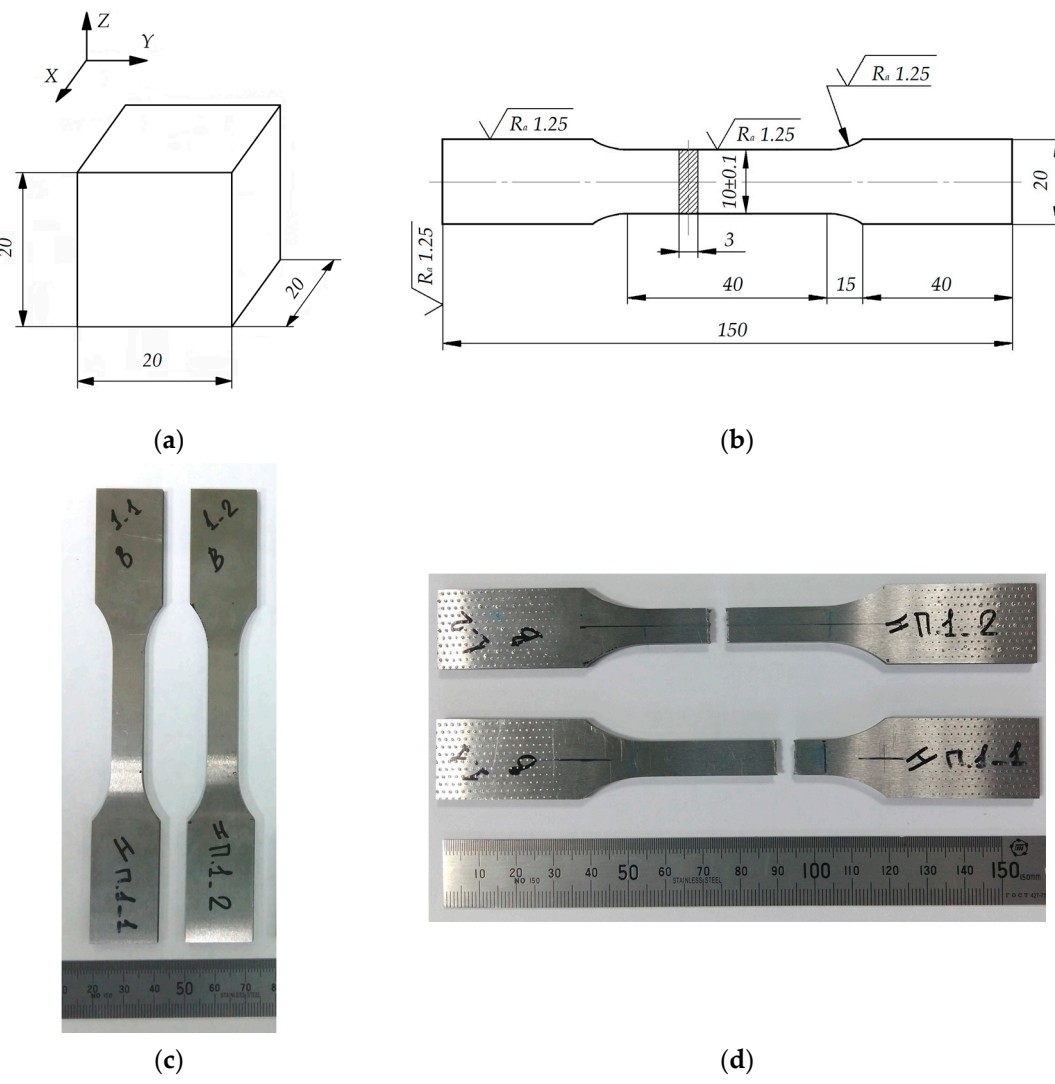

**Figure 2.** Specimens for tests: (**a**) schemes of specimens for hardness tests; (**b**) schemes of specimens for tensile tests; (**c**) general view of tensile test specimens; (**d**) destroyed specimens.

The quenching and tempering of the samples were produced in a chamber furnace CWF 12/23 with maximal temperature of 1200 °C and capacity of 23 L manufactured by Carbolite Gero (Hope Valley, UK).

The comparison of the hardness and wear resistance test results were done on the casted samples in the state of delivery according to TU 14-1-377-72 produced by LLC "Chelyabinsk Forging—Mechanical Plant" (Chelyabinsk, Russia).

## 2.3. Surface Morphology

The reduction of the roughness of the finished products was carried out by two methods of ultrasonic processing—cavitation-abrasive finishing and vibratory tumbling in water. It should be noted that dry vibratory tumbling is forbidden in many countries by sanitary norms and rules of production due to destructive action of the ceramic dust on human health (silicosis) [63].

Effects of a mechanical nature exert the most significant influence on the processing in liquid during ultrasonic cavitation abrasive finishing: cavitation, variable sound pressure, radiation pressure, acoustic streams of various scales, sound capillary effect. The introduction of ultrasonic vibrations into a liquid is an effective way of complex-shaped part processing with the internal cavities [44–46,64–66]. The method's effectiveness is due to many specific effects arising in a liquid technological medium under the influence of vibrations [67]. An acoustic pressure arises when ultrasound passes through a liquid as follows:

$$P_a = P_A \sin 2\pi f t, \tag{1}$$

where $P_A$ is a maximal amplitude of acoustic pressure (Pa); $f$ is oscillation frequency (Hz); $t$ refers to the propagating (collapse) time (s) [68,69]:

$$t = 0.915 R_{max} \left( \frac{\rho}{P_m} \right)^{\frac{1}{2}} \left( 1 + \frac{P_{vg}}{P_m} \right), \tag{2}$$

where $R_{max}$ is the radius of the cavity at the start of the collapse (m); $\rho$ is the medium density (kg·m$^3$); $P_m$ is the medium pressure at the time of collapse (Pa); or [70]

$$t = 0.915 R_{max} \left( \frac{\rho}{P_h} \right)^{\frac{1}{2}}, \tag{3}$$

where $P_h$ is the hydrostatic pressure surrounding the cavity (Pa). The maximum pressure developed in bubble $P_{max}$:

$$P_{max} = P_{vg} \left( \frac{P_m(\gamma - 1)}{P_{vg}} \right)^{\frac{\gamma}{(\gamma - 1)}}, \tag{4}$$

where $P_{vg}$ is pressure inside the cavity (of vapor-gas mixture) at maximum radius $R_{max}$ (in the bubble at its maximum size, pressure at the initial stage) (Pa); $\gamma$ is the polytropic index (exponent) for the gas mixture that is equal to the adiabatic exponent for the adiabatic process following Poisson's equation:

$$P \cdot V^\kappa = \text{const} \tag{5}$$

where $V$ is the volume (m$^3$), and $\kappa$:

$$\kappa = \frac{C_p}{C_v} \tag{6}$$

where $C_p$ and $C_v$ are heat capacity of gas at constant pressure and at constant volume, correspondingly. If:

$$\gamma = \frac{C - C_p}{C - C_v} \tag{7}$$

and $C$ is heat capacity of gas in the given process, then, for adiabatic process:

$$\gamma = \kappa \tag{8}$$

The polytropic exponent $\gamma$ determines the gas state in the cavity and is in the range of 1–1.33. The amplitude can be presented by acoustic force $F_A$ and system stiffness $k_a$:

$$A_m = \frac{F_A}{k_a} \tag{9}$$

At the same time, the stiffness of the system is determined by its mass:

$$K_a = 4\pi^2 \frac{m_a}{T^2}, \tag{10}$$

where $T$ is a period of natural oscillations [s]; or by cross-sectional area perpendicular to the line of the force application, Young's modulus and length of an element:

$$K_a = \frac{S_A \cdot E}{L}. \tag{11}$$

Because of the periodic action of tensile and compressive forces in the liquid, cavitation occurs. It consists of discontinuities in the continuity of the liquid with the subsequent collapse of these cavities. A feature of cavitation in the processing of solids is transforming a relatively low energy density of the sound field into a high density of local impulse action when cavitation bubbles collapse. Thus, if we consider the process of bubble collapse to be adiabatic, then the pressure inside is determined by expression:

$$P_m = P_{vg} \left( \frac{R_{max}}{R_{min}} \right)^{3\gamma} \tag{12}$$

or

$$P_m = P_{vg} \left( \frac{R_{max}}{R_{min}} \right)^{3-4} \tag{13}$$

where $R_{max}$ is a maximum bubble radius at the initial stage of collapse (m); $R_{min}$ is a minimum bubble radius at the end of the collapse (mm). The maximum pressure is determined by the ratio $\frac{R_{max}}{R_{min}}$. The results of classical studies [71–73] show that during the stretching period of the liquid $R_{max}$ exceeds the radius of the cavitation nucleus by 100–300 times, and the pressures $P_{max}$ can reach up to $10^7$–$10^{11}$ (Pa) at the stage of the collapse of the cavitation bubble, which causes plastic deformation of the solid surface and local destruction of the surface (erosion) when specific pressures are exceeded.

The second effect that determines the efficiency of ultrasonic liquid treatment is acoustic flows, the role of which is in the transfer and distribution of cavitation bubbles over the sound volume, which is especially important for the treatment of complex-profile surfaces. The nature of acoustic flows primarily depends on the mode of ultrasonic treatment, determined by the amplitude of oscillations of the end of the radiator $S_m$. Thus, large-scale acoustic flows are virtually absent at low-amplitude processing mode ($S_m < 10$–12 μm for water), and random sections of the sound volume are involved in cavitation. The transition to a high-amplitude mode ($S_m > 10$–12 μm) is abrupt and is explained by the strong absorption of acoustic energy during the development of the cavitation region at the end surface of the radiator [74], as a result of which directional hydrodynamic flows are formed, which carry out an active transfer of bubbles from the radiation surface to the treated surface. Formed flows lead to the formation of a stable cavitation area. The height of this area characterizes the depth of penetration of the cavitating liquid flow into the treated volume and depends on the amplitude of vibrations and the absorbing capacity of the process medium:

$$a = \frac{2\pi^2 f^2}{\rho c^3} \left( \frac{4}{3} \eta + \frac{(\gamma - 1)\theta}{\gamma C_v} \right), \tag{14}$$

where $\rho$ is medium density (kg·cm$^{-3}$); $c$ is the speed of sound in a medium (m·s$^{-1}$); $\eta$ is the viscosity of a fluid (Pa·s); $\theta$ is a coefficient of thermal conductivity of a material (W·(m·K)$^{-1}$); $C_v$ is the molar heat capacity at constant volume (J·(K·mol)$^{-1}$).

One of the methods for intensifying the solids' ultrasonic treatment is adding the abrasive powder to the working fluid—cavitation-abrasive finishing. The addition of insoluble abrasive particles to the sonicated liquid leads to a significant change in the processing. The presence of inhomogeneities in the technological liquid medium leads to a decrease in the liquid's cavitation strength and an increase in the number of cavitation centers, which increases the volume of the effective cavitation region. The mechanism of the effect of cavitation-abrasive finishing on the surface of the product is based as well on the micro-cutting action of abrasive particles, which acquire acceleration due to impulse transmission from shock waves' large-scale acoustic currents.

The ultrasonic cavitation-abrasive finishing was carried out on a half-wave magnetostrictive oscillatory system powered by a UZG 2.0/22 generator (JSK "Ultra-resonance", Yekaterinburg, Russia) (Figure 3a). An ultrasonic emitter of a rod three-half-wave magnetostrictive oscillatory system was immersed in water at a distance of 20 mm from the end face to the workpiece. The processing was carried out with the following parameters: the vibration frequency $f$ = 21,000 Hz, the vibration amplitude of the end face of the emitter $S_m$ = 20 μm, and the processing time $t$ = 120 s. During processing, the ultrasonic generator was operated in the automatic frequency control mode to maintain resonance conditions.

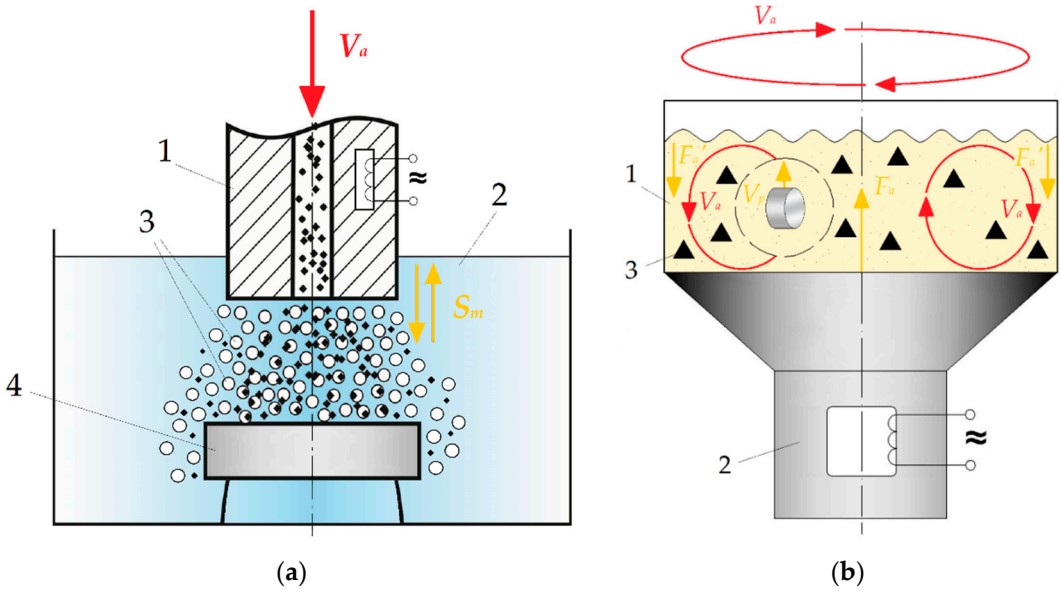

**Figure 3.** Schemes of ultrasonic treatment: (**a**) cavitation-abrasive finishing, where (**1**) is a transducer, (**2**) is medium, (**3**) is a cavitation bubble, (**4**) is a part to be processed, $V_a$ is a feed of abrasive, $S_a$ is a direction of induced oscillations; (**b**) vibration tumbling, where (**1**) is a tank with the parts, (**2**) is a transducer, (**3**) is an abrasive particle, $V_a$ is an abrasive movement speed, $V_p$ is a part movement speed, $F_a$ is an induced acoustic force, $F_a'$ is a medium force response.

The sample was placed in a radiator, on the bottom of which a layer of Elbor-R abrasive cubic boron nitride ($\beta$-BN) powder (JSC SPC "Abrasives and Grinding", Saint-Petersburg, Russia) of 2 mm thick was poured. An axial channel was made in the radiator to ensure the supply of abrasive powder (Elbor-R) to the cavitation zone. The optimal processing parameters were determined based on preliminary studies.

The vibratory tumbling was carried out while moving products and abrasive grains relative to each other in a vibrating container of an 80 L ZHM-80A vibratory tumbler finishing machine planetary drum type (Shengxiang, Zhejiang, China); a filler SCT VFC 10 mm × 10 mm ceramics prism gray (tumbling body) (CFT, Moscow, Russia) made of ceramics with 30% of silicon abrasive with a smooth

surface (Figure 3b). The processing was carried out in the following modes: a vibration frequency of 50 Hz, a filler weight of 50 kg, an operating time of 20.5 h, and an engine speed of 1440 rpm. The harmonic vibration amplitude was 6–7 mm.

*2.4. Characterization*

The dispersed (granulo- and morphometric) composition of the powders was determined on an optical particle shape analyzer 500NANO (Occhio, Liege, Belgium) using static image analysis according to ISO 13322-1: 2014. A scanning electron microscope VEGA 3 LMH 1,000,000× (Tescan, Brno, Czech Republic) determined particle morphology, real-time chemical analysis, and topology of the samples (with a surface inclination up to 30°).

An Empyrean diffractometer Series 2 (PANalytical, Almelo, The Netherlands) ranging from 25° to 70° carried out XRD measurements using monochromatic CuK$\alpha$ radiation ($\lambda$ = 1.5405981 Å) working at 60 kV and 30 mA in a step-scanning mode from 25° to 70° with a step size of 0.05° and a scan speed of 0.06°/min. The phase composition was analyzed using the PANalytical High Score Plus software and ICCD PDF-2 database. To conduct a quantitative phase analysis, the spectrum fitting method (Rietveld method) was used.

The microstructure and microrelief of the steel surface were studied using an Axio Observer D1m optical microscope (Carl Zeiss Microscopy GmbH, Kelsterbach, Germany) and a Phenom ProX scanning electron microscope following GOST 5639-82, GOST 1583-93 developed by the Euro-Asian Council for Standardization, Metrology, and Certification (EASC).

The measurements of the geometric parameters of the samples were carried out on a ScopeCheck MB multi-sensor coordinate measuring machine for high-precision measurements of large workpieces (Werth Messtechnik GmbH, Giessen, Germany).

The density of the samples $\rho$ was determined by hydrostatic weighing in distilled water using Archimedes' principle and compared with the theoretical values by an XP504 laboratory balance (Mettler Toledo, Columbus, OH, USA) with an accuracy of 0.001 g·cm$^{-3}$ [75,76].

Gas porosity was estimated on panoramic images with an area of 4 mm$^2$ (GOST 1583-93) using the Axio Observer D1m optical microscope; image processing including evaluation of the volume fraction of pores, the size distribution, and total porosity score was performed by a Thixomet Pro software (Thixomet, Saint Petersburg, Russia).

The following roughness parameters were evaluated during experiments according to EN ISO 4287:1997—arithmetic mean deviation ($R_a$), ten-point height ($R_z$), maximum peak-to-valley height ($R_{tm}$). The roughness was controlled by a high-precision profilometer, Hommel Tester T8000 (Jenoptik GmbH, Villingen-Schwenningen, Germany), by a Dektak XT stylus profilometer (Bruker Nano, Inc., Billerica, MA, USA) with a vertical accuracy of 5 Å (0.5 nm) and a tip radius of 12.5 µm, and by atomic force microscopy using a SMM-2000 multimicroscope (JSC "Plant PROTON" (MIET), Moscow, Zelenograd, Russia). Determination of submicro-roughness parameters was carried out by scanning by the constant height method, based on maintaining a constant distance between the cantilever and the sample.

All cross-sections of the samples were polished down to 1 µm using an ATM Machine Tools sampling equipment (ATM Machine Tools Ltd., Wokingham, UK).

Tensile tests under the application of static loads were carried out by a 5989 Universal Testing System (Instron ITW Company, Norwood, MA, USA) with a force of up to 600 kN at room temperature following GOST 1497-87. Three millimeter thick and ten millimeter width proportional flat specimens were produced for tests following GOST 11701-84.

A Wilson Tukon 2500 Automated Knoop/Vickers Hardness Tester (Instron ITW Company, Norwood, MA, USA) determined Vickers microhardness following GOST R ISO 6507-1-2007. A 574T Series Wilson Rockwell Hardness Tester (Instron ITW Company, Norwood, MA, USA) determined Rockwell's hardness following GOST 9013-59.

The wear resistance was determined by the method of water-jet wear by measuring the depth and width of the formed groove using a Calowear Instrument abrasion tester, which characterizes the resistance to abrasion of a surface (CSM Instruments, Peseux, Switzerland). The tests were carried out under the following conditions: a friction rate of 594 rpm, a static load of 0.25 N, a ball diameter of 25.4 mm, test time of 3, 6, 9 min, sample surface roughness parameter $R_a$ of 1.25 μm. An ODSCAD 6.2 measurement program (GFM LMI Technologies GmbH, Teltow, Germany) investigated the resulting wells' diameter and depth.

## 3. Results

### 3.1. Granulomorphometry and Samples' Production

The results of the granulo- and morphometric analysis of the powder showed that spherical particle shape, high fluidity, a microcrystalline structure, equiaxial morphology, and a small number of satellites characterize both of the steels (Figure 4, Table 3).

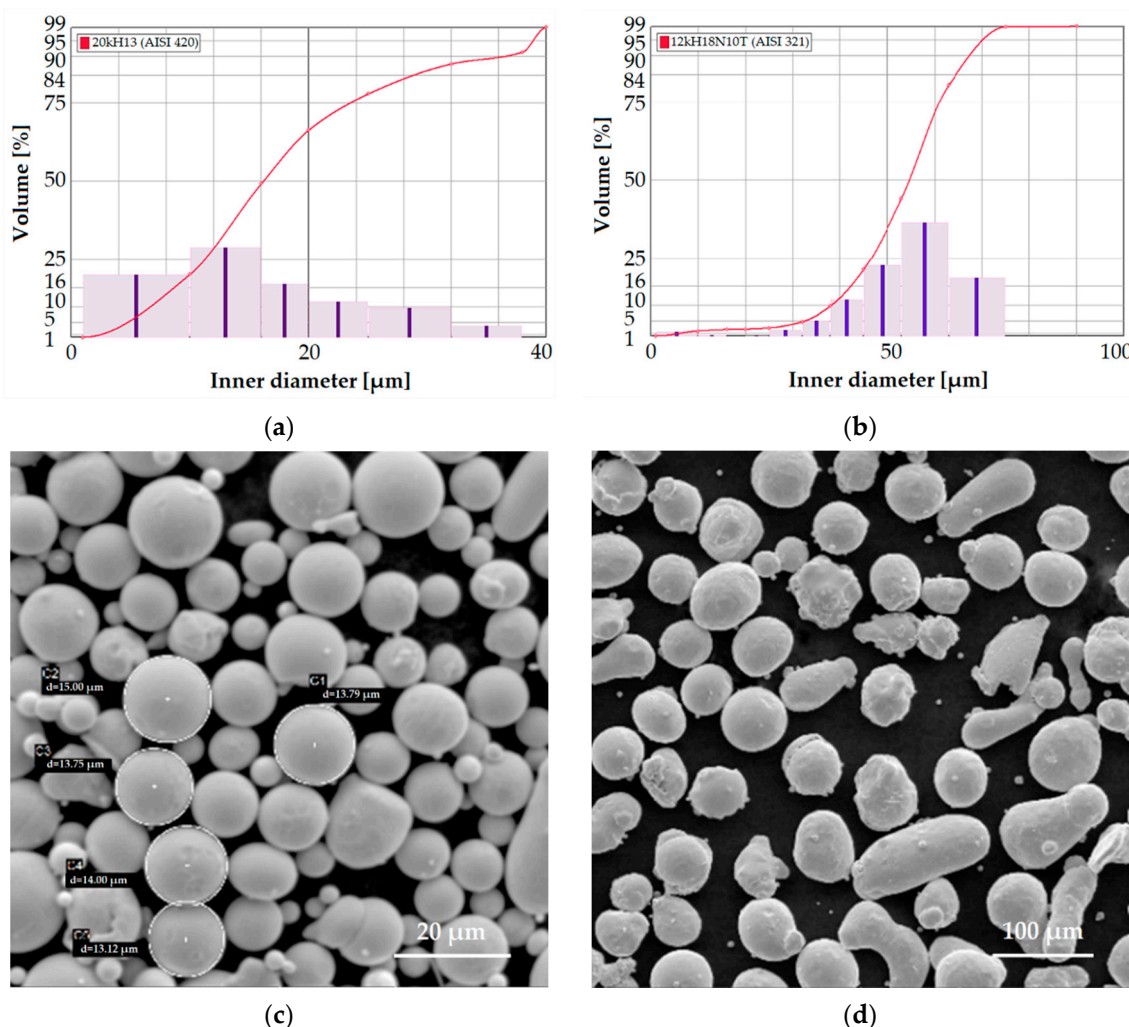

**Figure 4.** Granulometry and morphology (scanning electron microscopy) of the steel powders: (**a**) measurement results of PR 20kH13 (AISI 420); (**b**) measurement results of PR 12kH18N9T (AISI 321); (**c**) SEM-image of PR 20kH13 (AISI 420) particles, 2.98k×; (**d**) SEM-image of PR 12kH18N9T (AISI 321) particles, 510×.

**Table 3.** Shape parameters of the powders according to EN ISO 9276-6; 7; 8.

| Powder | Shape Factor $\varphi_i$ (Dimensionless) | Bluntness (%) | Roughness (mm) | Elongation (%) | Circularity (%) | Solidity (%) |
|---|---|---|---|---|---|---|
| PR 20kH13 (AISI 420) | 0–1 | 91.3 | 0.003 | 23.0 | 72.5 | 97.8 |
| PR 12kH18N9T (AISI 321) | | 73.9 | 0.031 | 22.0 | 65.2 | 90.6 |

Tracks were made from PR 20kH13 (AISI 420) and PR 12kH18N9T (AISI 321) steels at laser power in the range from 40 to 120 W, and scanning speeds from 50 to 150 mm·s$^{-1}$; the thickness of the powder layer applied to the substrate varied from 20 to 50 μm; the length of the track (melt pool) was 15 mm for all experiments. The height of the track above the substrate, the depth of the penetration zone of the underlying layer, the width of a single track, the width of the penetration zone, and the substrate's wetting angle in the zone of exposure laser radiation were analyzed for each experiment. The following laser powder bed fusion factors were selected for the manufacture of samples with the aim of further research because of the conducted analysis (Table 2):

- Layer thickness of 20 μm, laser power $P_l$ of 80 W, scanning speed vs. of 390 mm·s$^{-1}$ for PR 20kH13 (AISI 420) steel;
- Layer thickness of 20 μm, laser power $P_l$ of 100 W, scanning speed vs. of 100 mm·s$^{-1}$ for PR 12kH18N9T (AISI 321) steel.

*3.2. Microstructure*

The optical microscopy of the samples showed no visible cracks and pores. The microstructure of samples is shown in Figure 5. The grain structure and size are discerned at 1000× magnification (Figure 5b). The obtained microphotographs demonstrate paths of powder fusion by a laser beam. The average grain diameter of the observed area does not exceed 2 μm for 20kH13 (AISI 420) steel that corresponds to the 15th point (fine-grained group). However, it is not possible to identify individual phases in micrographs since they are very different from equilibrium conditions for the formation of the structure. The detected cells with an oriented structure characteristic of the laser powder bed fusion method are revealed at high magnifications (Figure 5a). The dendrites have no second-order axes in the microstructures of the produced samples. No non-metallic inclusions or traces of intergranular corrosion were found.

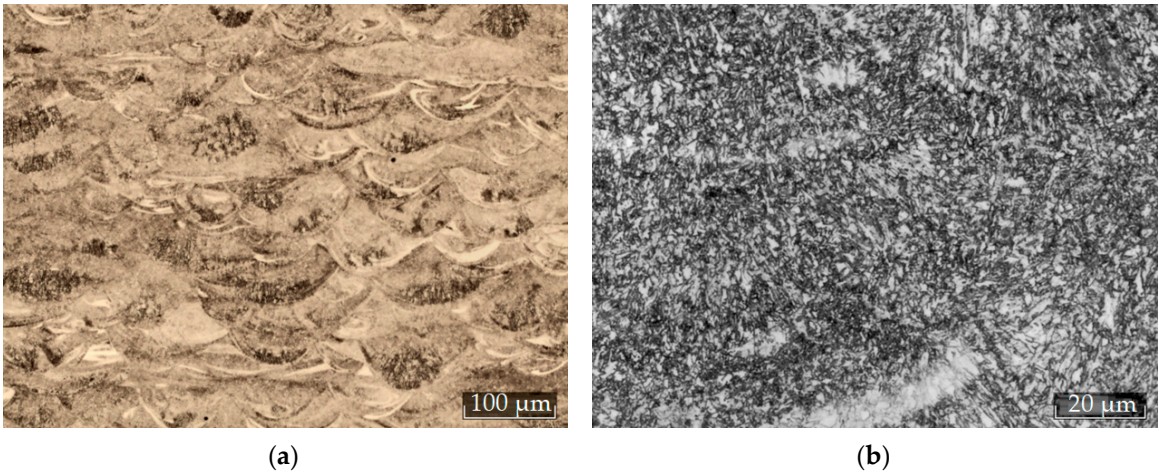

(**a**)        (**b**)

**Figure 5.** Optical microscopy of 20kH13 (AISI 420) steel sample after laser powder bed fusion: (**a**) 200×; (**b**) 1000×.

X-ray structural analysis showed the presence of a supersaturated solid solution of the $\alpha$-phase in the structure of the 20kH13 (AISI 420) sample and $\gamma$-phase in the 12kH18N9T (AISI 321) sample. The pore size does not exceed 0.01 mm that corresponds to point 3 by porosity point scale; the volume fraction of pores is in the range of 0.05%–0.1%.

### 3.3. Physical and Mechanical Properites

The obtained physical and mechanical characteristics of specimens made of 20kH13 (AISI 420) and 12kH18N9T (AISI 321) steels produced by the laser powder bed fusion method in comparison with the traditional manufacturing methods presented in Table 4.

**Table 4.** Physical and mechanical properties of 20kH13 (AISI 420) and 12kH18N9T (AISI 321) steel samples produced by laser powder bed fusion in comparison with the data obtained for traditionally produced samples.

| Material | Producing Method | Density ($\rho$) (g·cm$^{-3}$) | Young's Modulus (*E*) (GPa) | Yield Strength ($\sigma_{0.2}$) (MPa) | Tensile Strength ($\sigma_B$) (MPa) | Elastic Recovery ($\delta$) (%) |
|---|---|---|---|---|---|---|
| 20kH13 (AISI 420) | Laser powder bed fusion | 7.709 ± 0.004 | 208.918 ± 1.025 | 863.98 ± 7.38 | 1584.09 ± 3.58 | 7.31 ± 0.22 |
| | Cast + Quenching + low tempering | 7.7 [1,2] | - | 987 [1] | 1485 [1] | 9 [1] |
| | Cast + Quenching + high tempering | 7.7 [1,2] | 190–218 [2,3] | 635 [1] | 830 [1] | 10 [1] |
| 12kH18N9T (AISI 321) | Laser powder bed fusion | 7.905 ± 0.004 | 193.012 ± 1.018 | 576.02 ± 7.31 | 657.03 ± 3.32 | 48 ± 0.56 |
| | Cast + Quenching at 1050–1100 °C with cooling in water | 7.9 [1,2] | 175–195 [2,3] | 196 [1] | 540 [1] | 40 [1] |

[1] Provided according to GOST 5949-2018; [2] Provided according to GOST 5632-72; [3] the largest value corresponds to +20 °C and the smallest value corresponds to the +300 °C of working temperatures.

The results show that the mechanical characteristics of steels after laser powder bed fusion are higher or at the same level than those mentioned that are produced by traditional processing methods, which is explained by the features of the structure of steels obtained after the laser powder bed fusion method (supersaturation of the solid solution and mecodispersity of the structure) with almost equal density values.

### 3.4. Rougness

#### 3.4.1. 20kH13 (AISI 420) Anticorrosion Steel

The roughness parameter $R_a$ of the samples produced by laser powder bed fusion method from 20kH13 (AISI 420) steel was 7.24 ± 0.19 µm. The topology of the surface is shown in Figure 6. The measured results of roughness parameters ($R_a$, $R_z$, $R_{tm}$) are presented in Table 5; the obtained topology is shown in Figure 7 (visible area is 3.8 µm × 3.8 µm). The obtained profiles are presented in Figure 8. The untreated surface has a non-uniform topography with many inclusions formed due to metal splashing in the molten pool. The surface acquires a relatively regular structure and is characterized by the absence of inclusions after cavitation-abrasive finishing.

**Table 5.** The roughness parameters ($R_a$, $R_z$, $R_{tm}$) of the 20kH13 (AISI 420) steel sample surfaces before and after cavitation-abrasive finishing (after Gaussian regression).

| Method | Arithmetic Mean Deviation ($R_a$) (µm) | Ten-Point Height ($R_z$) (µm) | Maximum Peak-to-Valley Height ($R_{tm}$) (µm) |
|---|---|---|---|
| After laser powder bed fusion manufacturing | 7.24 | 72.9 | 83.7 |
| After ultrasonic cavitation abrasive finishing | 3.04 | 50.1 | 58.8 |

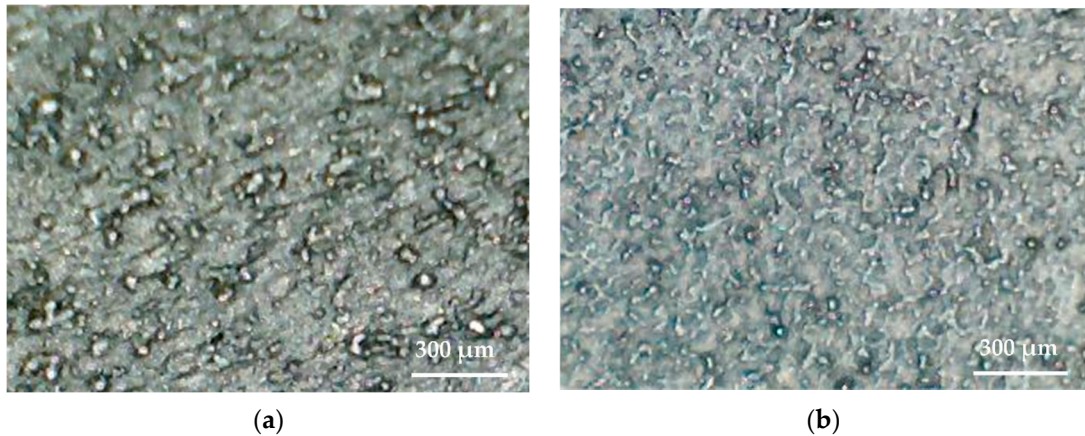

**Figure 6.** Surface topology of the samples (optical microscopy, 300×): (**a**) after production; (**b**) after cavitation-abrasive finishing.

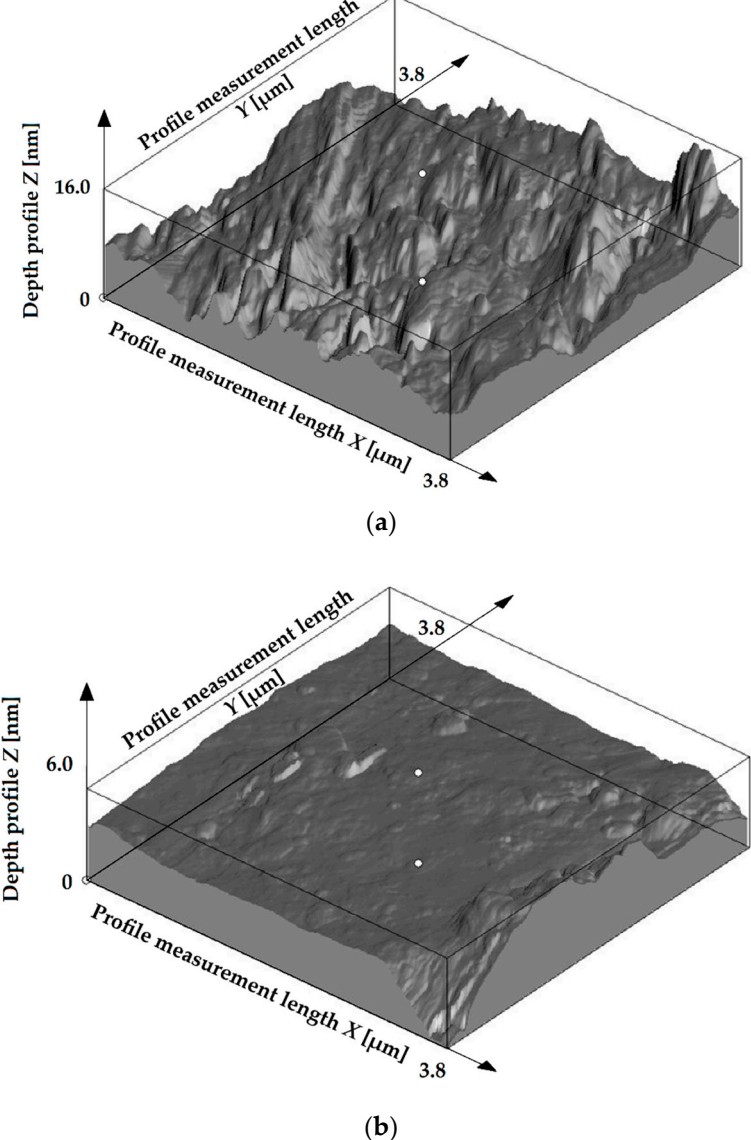

**Figure 7.** Surface topology of the samples (atomic force microscopy): (**a**) after production; (**b**) after cavitation-abrasive finishing.

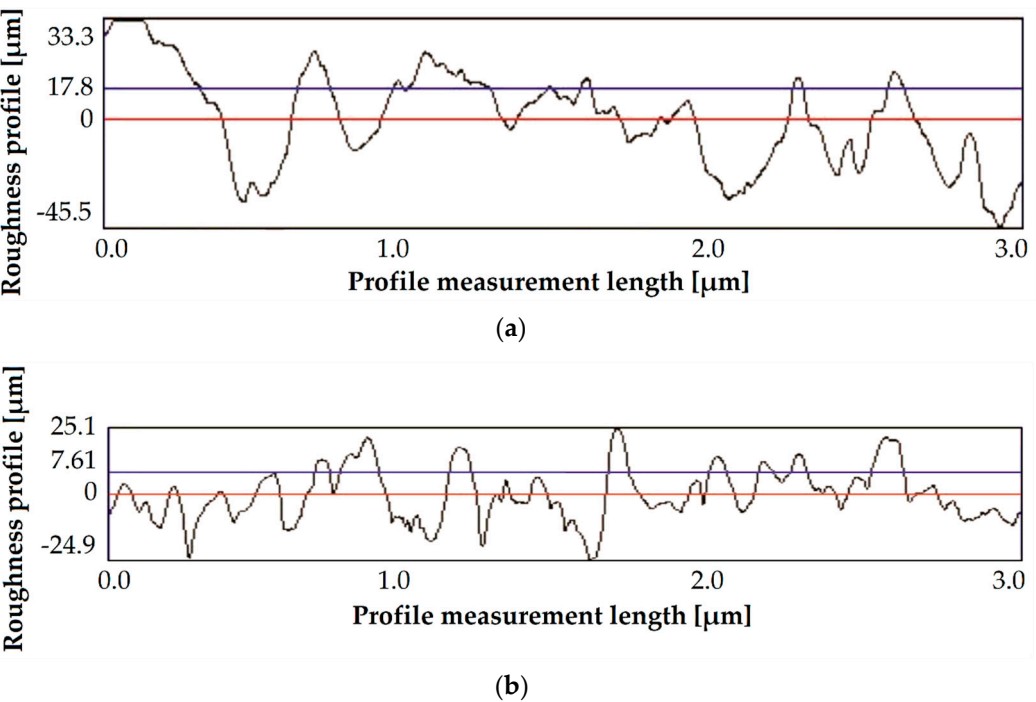

**Figure 8.** Surface profile of the samples: (**a**) after production; (**b**) after cavitation-abrasive finishing.

It is correlated to the distribution of cavitation activity, which depends on the treated surface's microgeometry parameters. Working fluids, which are cavitation bubbles and abrasive particles, have the most significant effect in places of most significant irregularities. In this case, the abrasive particles have damping functions, i.e., take on the energy arising from the collapse of cavitation bubbles, and the deformation of the surface is carried out due to the impact of the abrasive and not due to the effect of cumulative streams, which can lead to erosion. Thus, the applied cavitation-abrasive finishing is erosion-free and leads to surface smoothing.

It is quantitatively expressed in a decrease in the roughness parameters ($R_a$, $R_z$, $R_{tm}$) by 30%–60%, decreasing the average pitch of irregularities by more than two times. Increasing the processing time (over 120 s) does not increase the effect further.

### 3.4.2. 12kH18N9T (AISI 321) Anticorrosion Steel

The roughness parameter $R_a$ of the samples produced by laser powder bed fusion method from 12kH18N9T (AISI 321) steel varied in the range of 8.5–14.1 µm. The obtained topology is shown in Figure 9; measuring results are presented in Table 6. 3D-profiles of the surfaces are presented in Figure 10. The roughness parameter $R_a$ for the walls of after the vibratory tumbling was reduced by more than two times—from 14.1 to 5.0 µm.

**Table 6.** The roughness parameters ($R_a$, $R_z$) of the 12kH18N9T (AISI 321) steel surfaces before and after vibratory tumbling (before Gaussian regression).

| Method | Wall of the Sample | | Top of the Sample | |
|---|---|---|---|---|
| | Arithmetic Mean Deviation ($R_a$) (µm) | Ten-Point Height ($R_z$) (µm) | Arithmetic Mean Deviation ($R_a$) (µm) | Ten-Point Height ($R_z$) (µm) |
| After laser powder bed fusion manufacturing | 14.1 | 53.0 | 8.5 | 36.5 |
| After vibratory tumbling | 5.0 | 17.4 | 2.5 | 11.3 |

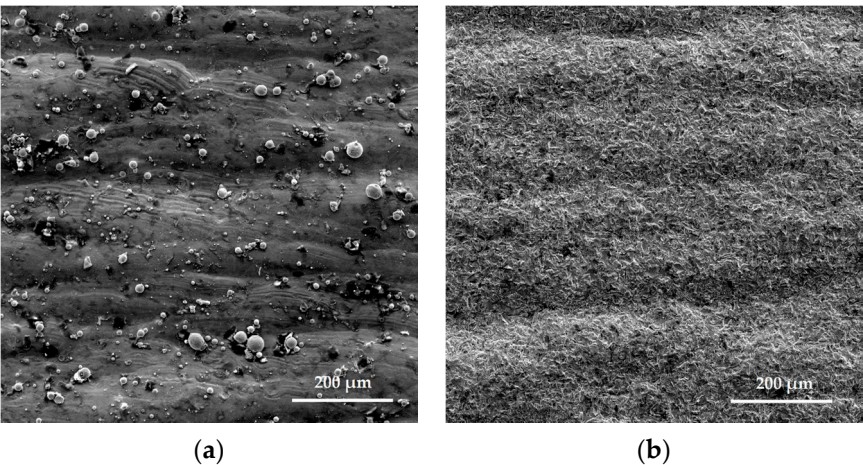

(**a**)  (**b**)

**Figure 9.** SEM-image in secondary electrons of the surface produced from 12kH18N9T (AISI 321) anticorrosion steel powder: (**a**) after laser powder bed fusion method, 500×; (**b**) after vibratory tumbling, 502×.

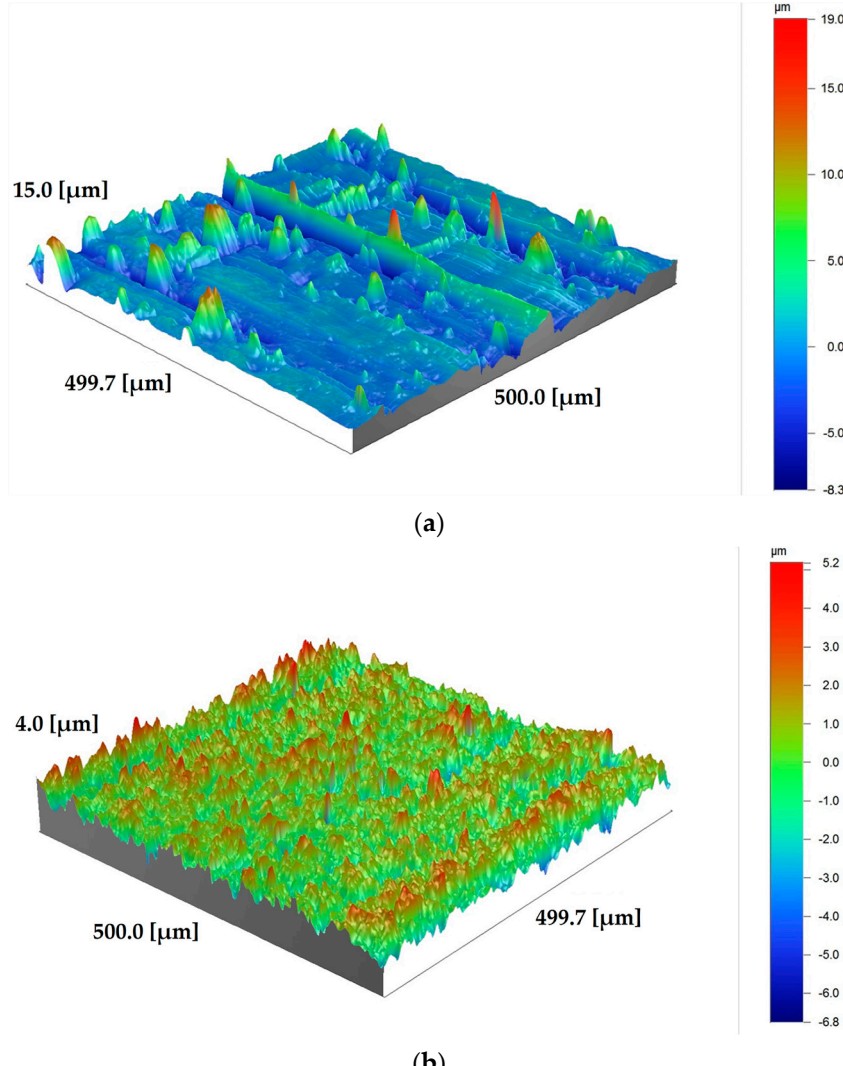

**Figure 10.** Surface topology of the samples (profilometry): (**a**) after production; (**b**) after vibration tumbling.

The surface longitudinal and traverse profiles on the area of 500 μm × 500 μm after Gaussian regression are presented in Figures 11 and 12. The measured roughness parameters $R_a$ and $R_z$ after regression are:

- 1.45 and 8.55 μm for raw sample after production;
- 1.25 and 7.05 μm for treated sample.

In the first case, the peaks of roughness profile (Figures 11b and 12b a peak of 12 μm) are associated with the presence of the unmelted granules on top of the surface that has metallurgical contact with the built sample. For the samples after vibratory tumbling, this type of morphology is absent; the surface retains a less pronounced wave character, without peaks, but with wells from the used abrasive, which are evenly distributed and regularized (Figure 11d).

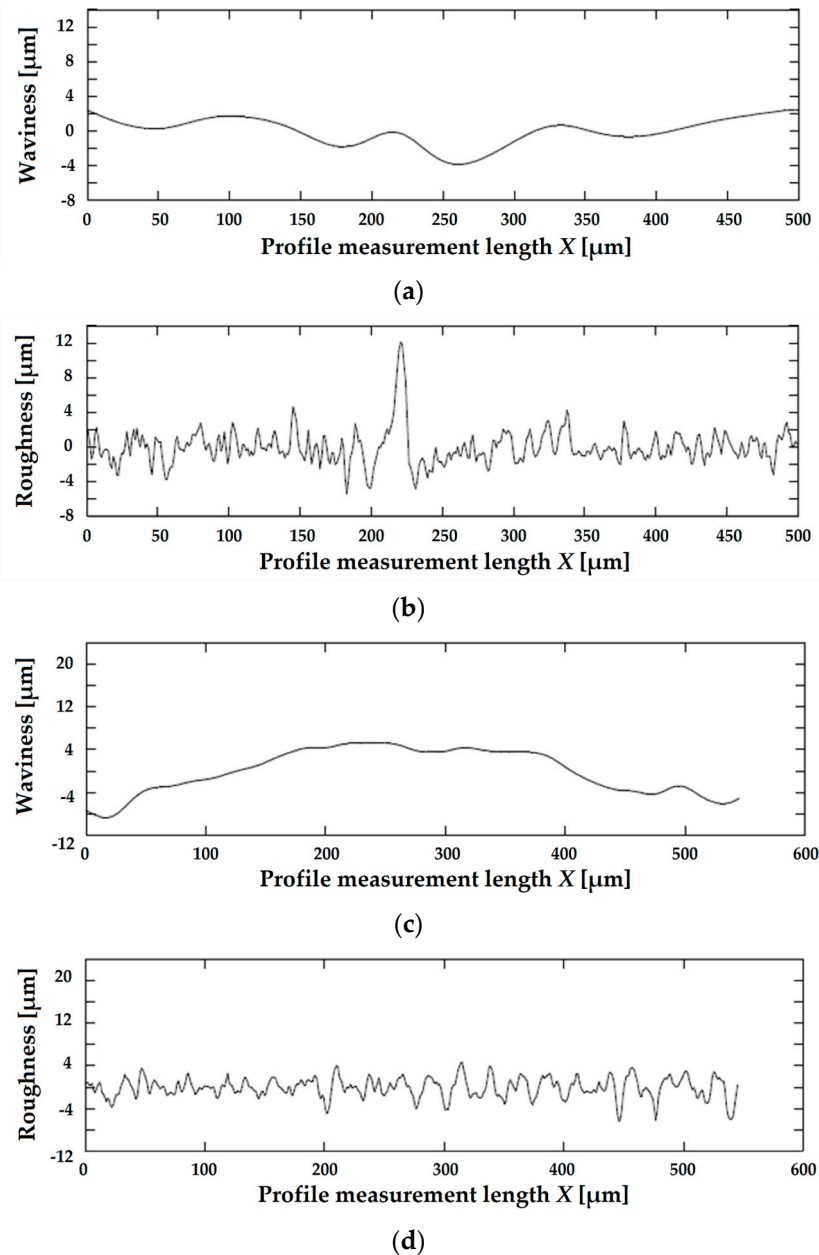

**Figure 11.** Longitudinal waviness and roughness profile after Gaussian regression (profilometry): (**a**) waviness after laser powder bed fusion method; (**b**) roughness profile after laser powder bed fusion method; (**c**) waviness after vibratory tumbling; (**d**) roughness profile after vibratory tumbling.

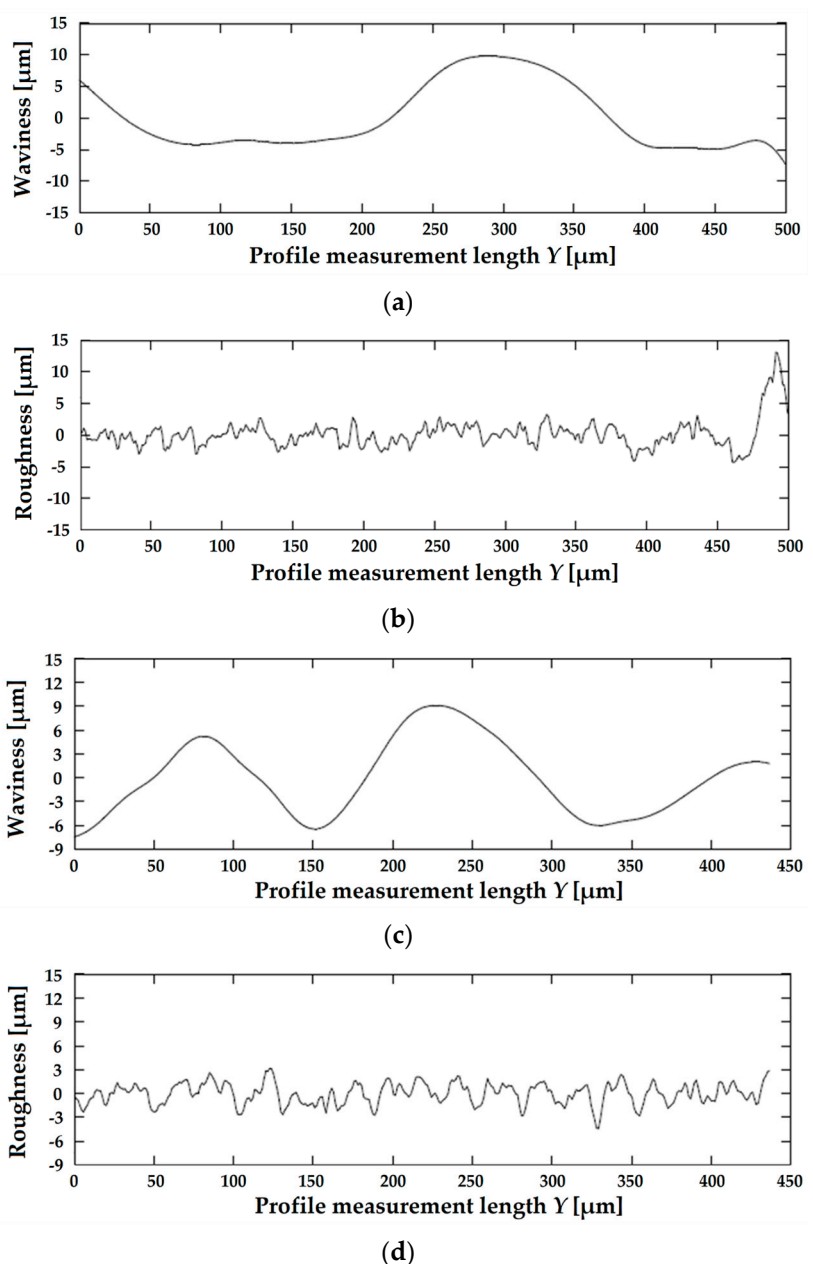

**Figure 12.** Transverse waviness and roughness profile after Gaussian regression (profilometry): (**a**) waviness after laser powder bed fusion method; (**b**) roughness profile after laser powder bed fusion method; (**c**) waviness after vibratory tumbling; (**d**) roughness profile after vibratory tumbling.

### 3.5. Heat Treatment, Hardness and Wear Resistance

The effect of subsequent heat treatment on the hardness of the 20kH13 (AISI 420) steel samples produced by laser powder bed fusion and traditional casting are presented in Table 7. The measured microhardness of the samples correlated to the data obtained for Rockwell hardness. The results of studies on water-jet wear of the laser powder bed fused samples and cast samples after combined quenching and low tempering are presented in Figures 13–15.

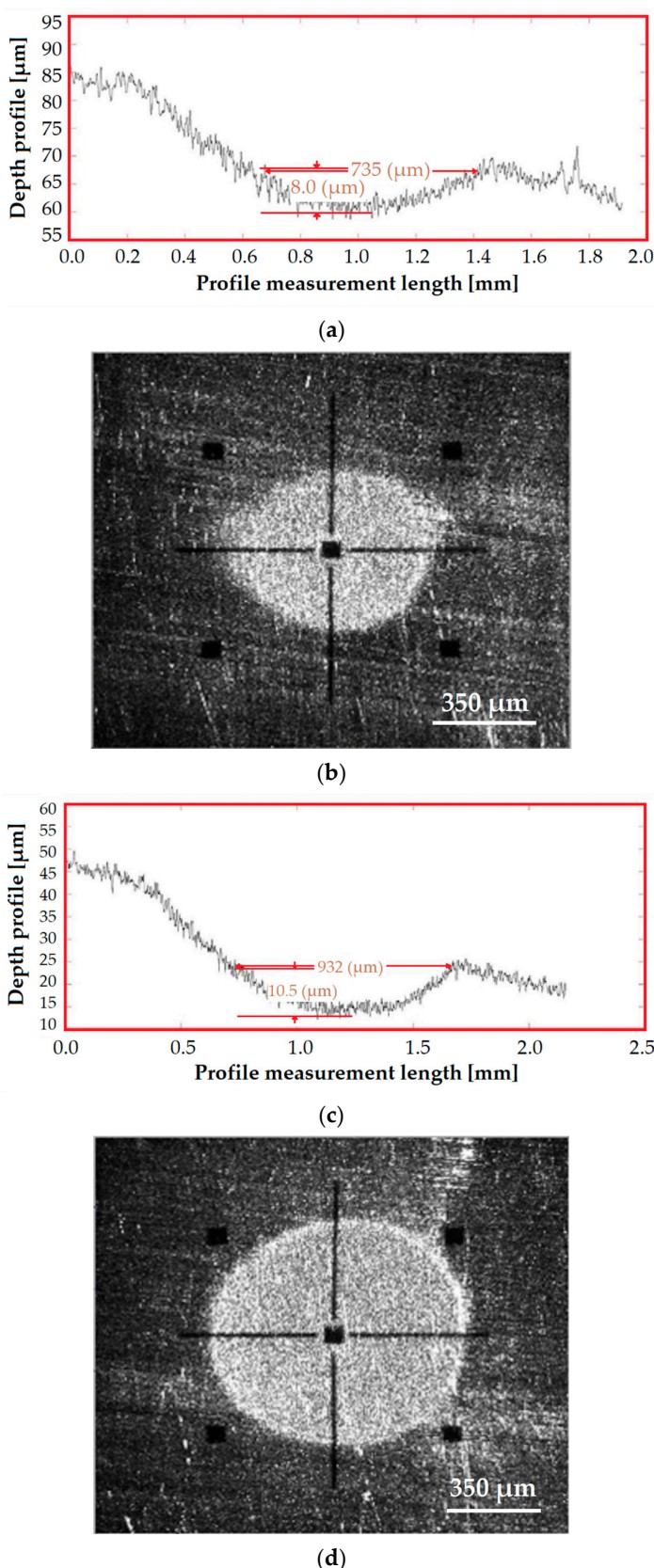

**Figure 13.** Profiles and microphotographs of the wells of the 20kH13 (AISI 420) steel samples after 3 min of test time (abrasion wear analyzer): (**a**) profile of the LPBF produced sample; (**b**) well image of the LPBF-produced sample; (**c**) profile of the cast sample treated by quenching with low tempering; (**d**) well image of the cast sample treated by quenching with low tempering.

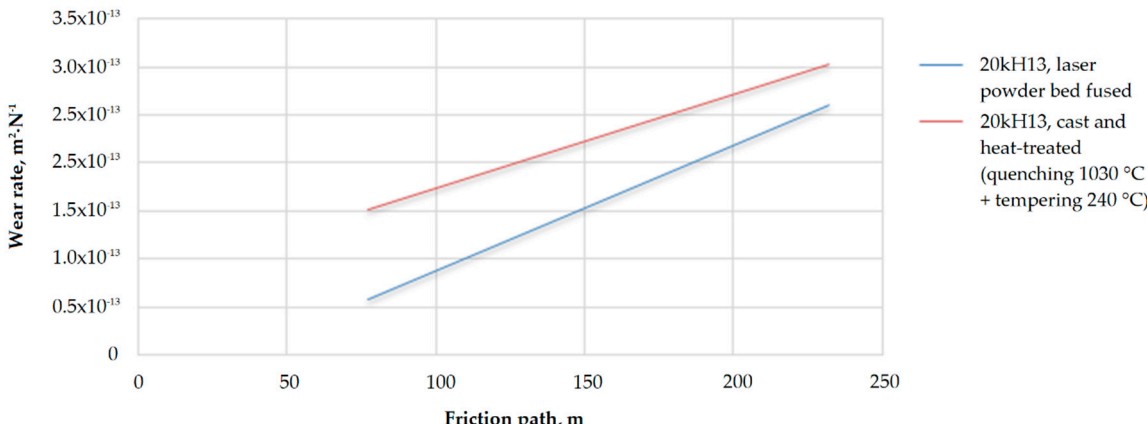

**Figure 14.** Dependence of the wear rate on the friction path of the 20kH13 (AISI 420) steel samples.

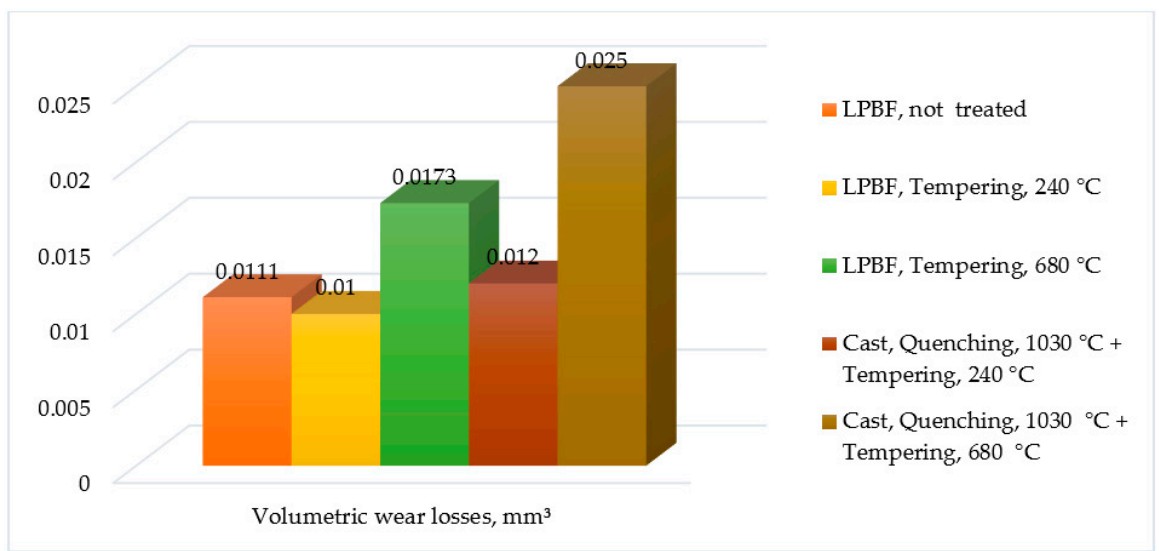

**Figure 15.** Histogram of volumetric losses during abrasive wear of steel 20kH13 (AISI 420), 9 min of wear time.

**Table 7.** Rockwell hardness of the samples made of the 20kH13 (AISI 420) steel in raw state after various types of post-processing.

| Producing Method | Hardening (°C) | Medium | Tempering (°C) | Medium | Rockwell Hardness HRC$_z$ [1] | Rockwell Hardness HRC$_{xy}$ [1] |
|---|---|---|---|---|---|---|
| Laser powder bed fusion | - | - | 240 | Air | 44.2 | 46.2 |
| | - | - | 680 | Oil | 38.7 | 39.1 |
| | - | - | Annealing, 760 | Air | 33.4 | 35.8 |
| | - | - | - | - | - | 46.25 |
| Cast | 1030 | Oil | 240 | Air | 43.0 | |
| | 1030 | Oil | 680 | Oil | 22.3 | |

[1] Provided for two different directions—along *Z*-axis and *X*- and *Y*-axes for the samples produced by laser powder bed fusion due to known anisotropy (orthotropy) of the properties and one value HRC for cast samples due to known isotropy of the properties.

## 4. Discussion

### 4.1. Estimation of the Obtained Effects

The provided study allows researching the possible steps of post-processing for the parts produced from structural anticorrosion chrome-nickel steels—20kH13 (AISI 420) and 12kH18N9T (AISI 321).

The samples' detected fine structure was obtained due to high cooling rates [18,77–79]—the liquid quenching mechanism is implemented at growing solids by the laser powder bed fusion method.

The presence of inhomogeneities in the technological liquid medium during ultrasonic cavitation abrasive finishing led to a decrease in the liquid's cavitation strength and an increase in a cavitation centers number [80]. The abrasive particles acquired acceleration because of the transfer of impulse energy from shock waves of acoustic micro- and macroflows [81]. It was expressed quantitatively in a decrease in the roughness parameters ($R_a$, $R_z$, $R_{tm}$) by 50%–60%, a two-fold reduction in the average pitch of irregularities, and an increase in the profile reference length by 10%. Practice showed that extended processing time, which exceeded 120 s, did not increase the obtained effect.

Besides, abrasive particles have damping functions and protected the surface from cavitation destruction. The surface had no traces of cavitation erosion; there were no grinding marks, which indicated a uniform effect over the entire processing area and intensive removal of the material layer. The layer change occured due to the destruction of the protrusions of the grain surface. The most significant effect was achieved by ultrasonic cavitation abrasive finishing (Table 5, Figure 8).

The analysis of the results (Table 7) shows that the hardness of 20kH13 (AISI 420) samples produced by the laser powder bed fusion method was higher than that of the cast and heat-treated 20kH13 (AISI 420) samples. The low tempering of the laser powder bed-fused samples led to a slight decrease in hardness. Given that low tempering reduces quenching stresses, this kind of heat treatment can be recommended for parts that work for wear, particularly for those chosen in this study part as a locking washer. The hardness after high tempering was significantly lower than after a low one, which was explained by the decomposition of martensite and its transition to sorbitol tempering.

Investigation of the effect of heat treatment on the wear resistance of 20kH13 (AISI 420) steel specimens produced by the laser powder bed fusion method showed that the wear resistance without subsequent heat treatment was higher than that of samples obtained by the traditional casting with special heat treatment (Figures 10 and 11). The abrasion resistance correlates with the measured hardness. The wear resistance decreases when the tempering temperature increases for all the samples under study. The values of the wear resistance of specimens obtained by the laser powder bed fusion method and by the laser powder bed fusion method with low tempering differ slightly. Simultaneously, they are slightly higher than the wear resistance of cast specimens after traditional quenching and low tempering.

## 4.2. Quantituve Evaluation of Cavitation-Abrasive Finishing Factors

Let us quantitatively evaluate the achieved results. The stiffness of the system for two types of steel samples produced by additive manufacturing will be:

$$K_{420} = 4.18 \cdot 10^9 \left[\frac{\text{N}}{\text{m}}\right], \tag{15}$$

$$K_{321} = 3.86 \cdot 10^9 \left[\frac{\text{N}}{\text{m}}\right]. \tag{16}$$

Then the period of the natural oscillations will be

$$T_{420} = \sqrt{4\pi^2 \frac{m_{420}}{k_{420}}} = \sqrt{4\pi^2 \frac{0.0616[\text{kg}]}{4.18 \cdot 10^9 \left[\frac{\text{N}}{\text{m}}\right]}} = 2.412 \cdot 10^{-5}[\text{s}], \tag{17}$$

$$T_{321} = \sqrt{4\pi^2 \frac{0.0632[\text{kg}]}{3.86 \cdot 10^9 \left[\frac{\text{N}}{\text{m}}\right]}} = 2.542 \cdot 10^{-5}[\text{s}], \tag{18}$$

when the period of forced oscillations is:

$$\tau = \frac{1}{f_{frc}} = 4.762 \cdot 10^{-5} [s].\tag{19}$$

The amplitude of the vibration can expressed as follows [82,83]:

$$\overline{A_m} = \overline{A_0} \cdot e^{\overline{\beta} \cdot \tau},\tag{20}$$

where $\overline{\beta}$ is the damping coefficient expressed as a complex number in the form $a + bi$:

$$\overline{\beta} = \overline{q} - \overline{\mu},\tag{21}$$

where $\overline{q}$ is the index of excited oscillations, and $\overline{\mu}$ is the coefficient of medium resistance, let us allow that:

$$\overline{\mu} \rightarrow 0\tag{22}$$

and $\overline{q}$ can be expressed as follows:

$$\overline{q} = \sqrt{\frac{\sqrt{h} - k}{2 \cdot m_n}},\tag{23}$$

where:

$$h = \frac{H_m}{H_0}\tag{24}$$

and

$$k = \frac{K_m}{K_0} = 1\tag{25}$$

where the values indexed $m$ are for the actual conditions and the values indexed 0 for conditions at the initial stage of processing. If the amplitude of the vibration $S_m$ is 20 μm transit to the sample in the ideal conditions, then:

$$\sqrt{h} = \sqrt{\frac{H_m}{H_0}} = 0.9995\tag{26}$$

and

$$\overline{\beta_{420}} = \overline{q_{420}} = |0.064|,\tag{27}$$

$$\overline{\beta_{321}} = \overline{q_{321}} = |0.063|.\tag{28}$$

The amplitude of the forced oscillations will be:

$$\overline{A_{420}} = \overline{A_0} \cdot e^{(|0.064| \cdot 4.76 \cdot 10^{-5})} = \overline{A_0} \cdot e^{3.05 \cdot 10^{-6}} = 1.000003 \cdot \overline{A_0} = 2.0 \cdot 10^{-6} [m],\tag{29}$$

$$\overline{A_{321}} = \overline{A_0} \cdot e^{(|0.063| \cdot 4.76 \cdot 10^{-5})} = \overline{A_0} \cdot e^{3.00 \cdot 10^{-6}} = 1.000003 \cdot \overline{A_0} = 2.0 \cdot 10^{-6} [m].\tag{30}$$

The applied force will be:

$$F_{A420} = K_{420} \cdot \delta = 4.18 \cdot 10^{9} \left[\frac{N}{m}\right] \cdot 6 \cdot 10^{-12} [m] = 25.08 \cdot 10^{-3} [N],\tag{31}$$

$$F_{A321} = K_{321} \cdot \delta = 3.86 \cdot 10^{9} \left[\frac{N}{m}\right] \cdot 6 \cdot 10^{-12} [m] = 23.16 \cdot 10^{-3} [N].\tag{32}$$

and acoustic pressure on 0.0004 m$^2$ of the sample:

$$P_{A420} = \frac{F_{A420}}{S_A} = \frac{25.08 \cdot 10^{-3} [N]}{0.0004 [m^2]} = 62.7 [Pa],\tag{33}$$

$$P_{A321} = \frac{F_{A321}}{S_A} = \frac{23.16 \cdot 10^{-3} [\text{N}]}{0.0004 [\text{m}^2]} = 57.9 [\text{Pa}]. \tag{34}$$

Thus, it should be noted that the period of the natural oscillations that are determined by the nature of the system was less than the chosen period (frequency) of forced oscillations. Self-oscillations resulting from the action of the internal energy of the system with a fixed frequency $f_{slf}$, close to the natural frequency $f_{nat}$, and a fixed amplitude; the reason, in this case, is associated with the low rigidity of the system and fluctuations in the acting force $F_A$; vibration frequency increases with increasing system rigidity (stiffness $K$) and decreases with decreasing workpiece thickness. It is recommended that the cavitation finishing be performed outside the free (natural) oscillation area to exclude the resonance phenomenon [84]:

$$0.7 < \frac{f_{frc}}{f_{slf}} < 1.3 \tag{35}$$

or

$$0.7 < \frac{T}{\tau} < 1.3 \tag{36}$$

then the process can be characterized as effective and stable. In the given case:

$$\frac{T_{420}}{\tau} = \frac{2.412 \cdot 10^{-5}}{4.762 \cdot 10^{-5}} = 0.51, \tag{37}$$

$$\frac{T_{321}}{\tau} = \frac{2.542 \cdot 10^{-5}}{4.762 \cdot 10^{-5}} = 0.53. \tag{38}$$

As it can be seen, the chosen frequency should be enlarged for further experiments to achieve a more effective and stable processing mode in cavitation finishing; however, the introduction of ceramic abrasive in the working zone changed the conditions in the working tank and allowed reduced frequency in combination with the effectiveness of abrasive granule deformation. For improving parameters of processing up to the ratio of stable processing without adding additives that influence medium resistance, the frequency of the forced oscillations should be in the interval of 29–54 kHz for 20kH13 (AISI 420) steel and 28–52 kHz for 12kH18N9T (AISI 321) steel. If the amplitude of the forced vibrations in the working zone is small and measures in micrometers, then the difference $\delta$ between $\overline{A_0}$ and $\overline{A_m}$ is extremely small and is measured in picometers, which 0.01 of an angstrom (Å). With the known acoustic pressure, it is possible to evaluate the acoustic pressure's amplitude in each moment of the cycle. At the same time, it should be noted that frequency higher than 30 kHz and up to 1 MHz could be harmful to the biological process in the human body since arising cavitation with bubble formation with a diameter less than 1 μm (ultrasound surgery) [85], when ultrasound in the range 2–29 MHz is used in echography; the works should be conducted according to the sanitary norms and rules of production.

## 5. Conclusions

### 5.1. Structure and Surface Quality

In the process of the laser powder bed fusion method in the steels under study, as a result of high cooling rates, a finely dispersed structure with supersaturated solid solutions is formed, which provides higher strength characteristics of steels after additive manufacturing than after traditional processing methods.

The maximum effect on the change in surface quality was achieved by ultrasonic cavitation-abrasive finishing based on the combination of cavitation, acoustic flows, and abrasive particles that were evaluated quantitatively. A twofold decrease in the parameters of micro- and submicroroughness was achieved (roughness parameter $R_a$ was reduced from 7.24 to 3.04 μm), and no erosion caverns were detected on the researched surfaces. The use of a vibratory grinder reduces the roughness parameter

$R_a$ of the sample walls made of 12kH18N9T (AISI 321) powder from 14.1 to 5 μm. However, it should be borne in mind that using a vibratory grinder usually reflects the service life of parts negatively.

The quantitative evaluation showed that the difference in the chosen material's mechanical and physical properties did not reflect the maximal amplitude of $2.0 \cdot 10^{-6}$ m when δ of the samples was about $6 \cdot 10^{-12}$ m. The chosen frequency ratio to the natural frequency was about 0.5 and can be improved to improve processing effectiveness and stability without adding abrasive. The frequency of the forced oscillations should be in the interval of 29–54 kHz for 20kH13 (AISI 420) steel and 28–52 kHz for 12kH18N9T (AISI 321) steel.

### 5.2. Resistance to Abrasion Wear

The obtained abrasion resistance correlated with the measured hardness values. The wear resistance decreased with an increase in the tempering temperature for all the samples—the volume of worn material increased by 1.5–2 times for samples with a tempering temperature of 680 °C] in comparison with the raw part or after tempering at 240 °C. The values of wear resistance and hardness of additively manufactured specimens and the specimens that were additionally subjected to low tempering differ insignificantly (HRC of 46.25 and 46.2, correspondingly). However, they are slightly higher than the wear resistance of samples after traditional hardening and low tempering (HRC of 45.9). Considering that low tempering reduces residual stresses, heat treatment can be recommended for additively manufactured parts working for wear.

Analysis of the research results shows that, for complex-shaped parts made of corrosion-resistant steels by the laser powder bed fusion, it is promising to use heat treatment to improve mechanical characteristics and ultrasonic treatment in order to improve the surface quality in post-processing.

**Author Contributions:** Conceptualization, S.N.G. and M.A.V.; methodology, T.V.T. and S.K.S.; software, Y.A.M. and P.A.P.; validation, A.S.M. and S.K.S.; formal analysis, A.S.M. and A.A.F.; investigation, A.A.F., P.A.P. and Y.A.M.; resources, M.A.V., T.V.T.; data curation, A.A.O. and Y.A.M.; writing—original draft preparation, A.A.F. and S.K.S.; writing—review and editing, T.V.T. and A.A.O.; visualization, A.A.O. and P.A.P.; supervision, S.N.G.; project administration, S.N.G. and M.A.V.; funding acquisition, A.S.M. All authors have read and agreed to the published version of the manuscript.

**Funding:** This research was funded by the Russian Science Foundation, grant number No. 20-19-00620.

**Acknowledgments:** The research was done at the Department of High-Efficiency Processing Technologies of MSTU Stankin.

**Conflicts of Interest:** The authors declare no conflict of interest. The funders had no role in the design of the study; in the collection, analyses, or interpretation of data; in the writing of the manuscript, or in the decision to publish the results.

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
