# Peer review of "Effect of Cavitation Erosion Wear, Vibration Tumbling, and Heat Treatment on Additively Manufactured Surface Quality and Properties"

_metals, doi:10.3390/met10111540_

Round 1
Reviewer 1 Report
The modifications from the original submission are satisfactory.
Author Response
Dear Reviewer,
We would like to thank you for your kind evaluation of our work and wish you much success in your researches and results.
Best regards,
Authors
Reviewer 2 Report
The authors aim to investigate the Effect of cavitation erosion wear and vibration tumbling on additively manufactured parts' surface quality. The topic is interesting. However, the manuscript is redundant, including too many contents which are not closely related to the key focus of this manuscript. The manuscript should be thoroughly revised and the redundant contents should be removed. Too many redundant sentences are currently included in the introduction section. The authors should focus more on the surface quality. The conclusion section should be simplified. Only the new findings related to the surface quality should be remained. “in the interval of 29÷54 kHz for 20kH13 (AISI 420) steel and 28÷52 kHz 554 for 12kH18N9T (AISI 321) steel”, “÷” has been used for many times in this manuscript. However, it is very strange to use this symbol at this situation.Author Response
Response to Reviewer 2 Comments
Dear reviewer,
Thank you very much for your kind evaluation of our work. We do agree with all your proposals and comments and have modified the manuscript according to them.
Introduced changes were marked by green in the text of the manuscript.
Kind regards,
Authors.
Point 1: However, the manuscript is redundant, including too many contents which are not closely related to the key focus of this manuscript. The manuscript should be thoroughly revised and the redundant contents should be removed. Too many redundant sentences are currently included in the introduction section.
Response 1: Thank you for your valuable comment. We have tried to revise our abstract and introduction according to your suggestions. We hope that you find the introduced changes valuable and more focused on the research aim.
Point 2: The conclusion section should be simplified. Only the new findings related to the surface quality should be remained.
Response 2: Thank you for this suggestion; we have tried to simplify our conclusions: shortened them as far as possible without losing the main idea and divided them into two sections.
Point 3: “in the interval of 29÷54 kHz for 20kH13 (AISI 420) steel and 28÷52 kHz 554 for 12kH18N9T (AISI 321) steel”, “÷” has been used for many times in this manuscript. However, it is very strange to use this symbol at this situation.
Response 3: It is an international symbol of interval for [1]. The correct name is Obelus [2]. We have tried to replace it with a dash, but for us, it looks quite confusing in some fragments as it can be read as the minus sign:
- Sm > 10 – 12 [μm] (all the examples are provided in the file)
We have published before the papers in MDPI with the obelus and it was never a problem [3].
Reference:
- Writing Systems and Punctuation. The Unicode® Standard: Version 10.0 – Core Specification (PDF). Unicode Consortium. June 2017. p. 280, Obelus.
- https://en.wikipedia.org/wiki/Obelus
- Metel, A.S.; Stebulyanin, M.M.; Fedorov, S.V.; Okunkova, A.A. Power Density Distribution for Laser Additive Manufacturing (SLM): Potential, Fundamentals and Advanced Applications. Technologies2019, 7, 5. DOI: 10.3390/technologies7010005

Reviewer 3 Report
I have following recommendations:
- 2. Materials and Methods – add to Materials section (from Introduction, line 133 - 138). Indicate the designation of materials according to EN ISO and chemical composition.
- 2.2 Indicate the standard for the evaluation of roughness according EN ISO and the parameters that were measured in the experiment
- Table 2 – for quantity add units
- If it is roughness parameter, in the text and in the tables replace „roughness“ words „roughness parameter“, for example: do not use „roughness Ra“, but „roughness parameter Ra“
- always list units in square brackets, for example [MPa], [mm].
Fig. 7 – improve readability – add scale and the numbering of the axes x, y, z.
Fig. 11a) 11c) 12a) 12c) – correct axis names: axis x – profile measurement length [mm], axis y – waviness [mm]
Fig. 13 a) c) correct axis names: axis x – profile measurement length [mm], axis y – depth profile [mm]
Fig. 13 b) d) improve readability – add scale
Author Response
Response to Reviewer 3 Comments
Dear reviewer,
Thank you very much for your kind evaluation of our work. We do agree with all your proposals and comments and have modified the manuscript according to them.
Introduced changes were marked by yellow and red in the text of the manuscript.
Kind regards,
Authors.
Point 1: - 2. Materials and Methods – add to Materials section (from Introduction, line 133 - 138). Indicate the designation of materials according to EN ISO and chemical composition.
Response 1: Thank you for your valuable suggestion. We have moved the mentioned fragment to the section Materials and Methods and have tried to indicate materials by EN ISO. The chemical composition was also provided. We hope that we got the reviewer's idea and, if the reviewer would like us to modify the designation of the materials from AISI to EN ISO throughout the manuscript, we are ready to do.
Point 2: - 2.2 Indicate the standard for the evaluation of roughness according EN ISO and the parameters that were measured in the experiment.
Response 2: Thank you for your valuable comment. All the roughness parameters are indicated according to EN ISO 4287:1997 throughout the manuscript.
Point 3: - Table 2 – for quantity add units
Response 3: All the units are added according to EN ISO 9276-6; 7; 8.
Point 4: - If it is roughness parameter, in the text and in the tables replace „roughness“ words „roughness parameter“, for example: do not use „roughness Ra“, but „roughness parameter Ra“
Response 4: It was revised throughout the manuscript and marked yellow.
Point 5: - always list units in square brackets, for example [MPa], [mm].
Response 5: It was revised throughout the manuscript and marked yellow and red. In some places, we found it quite confusing; we hope that the editors will help us to revise it following the journal standards on the stage of proofreading.
Point 6: Fig. 7 – improve readability – add scale and the numbering of the axes x, y, z.
Response 6: Figure 7 is revised.
Point 7: Fig. 11a) 11c) 12a) 12c) – correct axis names: axis x – profile measurement length [mm], axis y – waviness [mm]
Response 7: All the subfigures of Figures 11 and 12 are revised.
Point 8: Fig. 13 a) c) correct axis names: axis x – profile measurement length [mm], axis y – depth profile [mm]
Response 8: Subfigures 13a, c are revised.
Point 9: Fig. 13 b) d) improve readability – add scale.
Response 9: Subfigures 13b, d are revised.

Reviewer 4 Report
This paper introduced two types of post-processing methods – ultrasonic cavitation abrasive finishing and vibration tumbling and discussed their influence on the physical, mechanical properties and surface quality of the LPBF samples. The paper has been improved after first round of revision and can be accepted for publication after answering the following questions: 1. Line 285, “6-7 mm” or “6÷7 mm” ? This kind of punctuation should be checked throughout the paper. 2. Line 366, where is the X-ray results? Any evidence? figure? 3. The characterization of powder shape parameters, such as Bluntness, Roughness, and Elongation using the optical particle shape analyzer has not been well accepted. The authors are encouraged to cite the following related papers in order to make their statement more convinced: (1) Characterization of spherical AlSi10Mg powder produced by double-nozzle gas atomization using different parameters. Trans. Nonferrous Met. Soc. China 29(2019) 374−384. (2) Characterization of Ti6Al4V powders produced by different methods for selective electron beam melting, J. Min. Metall. Sect. B-Metall. 55 (1) B (2019) 121-128. 4. A scale bar should be added to Figure 6. 5. For Figure 7, it would be better to use a colorful figure (bar) like Figure 10 to show the roughness. 6. Have the authors done the tensile test for the samples after ultrasonic cavitation abrasive finishing and vibration tumbling treatment? If yes, any improvement?Author Response
Response to Reviewer 4 Comments
Dear reviewer,
Thank you very much for your kind evaluation of our work. We do agree with all your proposals and comments and have modified the manuscript according to them.
Introduced changes were marked by the green (one place) and blue in the text of the manuscript.
Kind regards,
Authors.
Point 1: The paper has been improved after first round of revision and can be accepted for publication after answering the following questions: 1. Line 285, “6-7 mm” or “6÷7 mm” ? This kind of punctuation should be checked throughout the paper.
Response 1: It is an international symbol of interval for [1]. The correct name is Obelus [2]. We have tried to replace it with a dash, but for us, it looks quite confusing in some fragments as it can be read as the minus sign:
- Sm > 10 – 12 [μm] (provided in the file)
We have published before the papers in MDPI with the obelus and it was never a problem [3]. Marked green in the text.
Reference:
- Writing Systems and Punctuation. The Unicode® Standard: Version 10.0 – Core Specification (PDF). Unicode Consortium. June 2017. p. 280, Obelus.
- https://en.wikipedia.org/wiki/Obelus
- Metel, A.S.; Stebulyanin, M.M.; Fedorov, S.V.; Okunkova, A.A. Power Density Distribution for Laser Additive Manufacturing (SLM): Potential, Fundamentals and Advanced Applications. Technologies2019, 7, 5. DOI: 10.3390/technologies7010005.
Point 2: Line 366, where is the X-ray results? Any evidence? figure?
Response 2: The following lines 358-361 show the results:
X-ray structural analysis showed the presence of a supersaturated solid solution of the α-phase in the structure of the 20kH13 (AISI 420) sample and γ-phase in the 12kH18N9T (AISI 321) sample. The pore size does not exceed 0.01 [mm] (point 3); the volume fraction of pores is in the range of 0.05 – 0.1 [%].
X-ray structural analysis did not show anything particular and unusual; thus, we have decided not to show the relevant figure since there are more than enough figures in the paper.
We have also added data on the used quantitative method (Lines 284-286):
The phase composition was analyzed using the PANalytical High Score Plus software and ICCD PDF-2 database. To conduct a quantitative phase analysis, the spectrum fitting method (Rietveld method) was used.
Point 3: The characterization of powder shape parameters, such as Bluntness, Roughness, and Elongation using the optical particle shape analyzer has not been well accepted. The authors are encouraged to cite the following related papers in order to make their statement more convinced: (1) Characterization of spherical AlSi10Mg powder produced by double-nozzle gas atomization using different parameters. Trans. Nonferrous Met. Soc. China 29(2019) 374−384. (2) Characterization of Ti6Al4V powders produced by different methods for selective electron beam melting, J. Min. Metall. Sect. B-Metall. 55 (1) B (2019) 121-128.
Response 3: We have added measuring units of the used analyzer according to EN ISO 9276-6; 7; 8. We should that to cite the reviewer's papers is quite an ethical issue that is not OK for us, especially when it has no relation to steels or laser additive manufacturing. However, we have provided both of the references in the paper.
Point 4: A scale bar should be added to Figure 6.
Response 4: A scale bar is added.
Point 5: For Figure 7, it would be better to use a colorful figure (bar) like Figure 10 to show the roughness.
Response 5: It is not possible to use a colorful figure for this type of atomic force microscope. The relevant data are added to the figure title.
Point 6: Have the authors done the tensile test for the samples after ultrasonic cavitation abrasive finishing and vibration tumbling treatment? If yes, any improvement?
Response 6: With regard to tensile tests after vibratory tumbling and ultrasonic cavitation abrasive finishing, testing after these types of processing was carried out directly on the parts under conditions as close as possible to operational ones.

This manuscript is a resubmission of an earlier submission. The following is a list of the peer review reports and author responses from that submission.
Round 1
Reviewer 1 Report
The manuscript compares the effects of innovative post-processing techniques on parts produced by selective laser melting in comparison to conventional cast material.
The manuscript is generally well organized and all experimental procedures clearly described. The evaluation of the effects of the post-processing was carried out on two real parts for aerospace applications. Despite the experimental approach is quite practical focusing on two real aerospace application, results are of industrial interest.
Quality of some figures can be improved: Figure 7, figure 10
Reviewer 2 Report
The paper is about the impact of surface finish of the SLMed part after fabrication on its Physical, Mechanical properties of the parts for aerospace application. It is an interesting job, but there are still some comments before its publish.
1.For abstract section, in line 20, it would be better to add more details regarding the strength characteristics, how much higher, can you present the increase in percentage?
Line 28, it would be better to add a final sentence to give an overall potential of this study for the intended application.
- In line 50, can you explain why that is? Why the surface quality of additive manufactured part is low? For instance : Is it due to the partially melted particles?
- The information in section 2.1 from line 133 to 164 can be transferred in to introduction section as an explanation of why steel was selected for this study and what are the requirements for final product?
- line 165, the dimensions of the 3D printed samples would be good to be added. For hardness test what was the dimensions of the samples and how many samples were tested? Also the images of the printed samples would be good to be added
- For tensile test samples how many samples were tested and what was the orientation of 3D printing?
- Line 174, in table 1, can you please justify the parameters? Are they optimised parameters from any previous studies?
- It would be good to combine figure 2a and figure 7 to show that the part was successfully printed according to its original design and STL files
- And figure 2b and figure 9 can be combined to show that the part was 3D printed according to its original geometry and design
- Figure 8: the scale bar needs to be improved and more clear
Reviewer 3 Report
This paper describes the effect of two kinds of polishing process of the ultrasonic washing and vibratory tumbling in water on the metal parts surfaces fabricated by a laser additive manufacturing. Laser additive manufacturing has a demerit that the fabricated parts surface is generally rough because the process is based on the melting process of metal powder by intense laser irradiation. In the technical point of view, some useful results are obtained. However, from an academic point of view, a novelty is difficult to be found through the manuscript. In addition, most of the experimental results can be explained by previous other papers. Thus, this paper is not suitable for this journal. Metals, and might be acceptable as a 'Technical Paper' in an appropriate other journal.
The other revised points for further submission to other journal are indicated for reference as follows:
1) 1. Introduction: the story is redundant.
2) p.4, L143: Figure1→Figure2 ?
3) p.7, L265: A tests were→Tests were ?
4) p.13, L357: By other words→In other words?
5) 4. Discussion: Discussion is poor; experimental results are just explained in this section. A newly obtained result cannot be found.